# Managing Nanomaterials in the Workplace by Using the Control Banding Approach [note 1]

**DOI:** 10.3390/ijerph20116011

**Published:** 2023-05-31

**Authors:** Delfina Ramos, Luis Almeida

**Affiliations:** 1ISEP—School of Engineering, Polytechnic of Porto, 4249-015 Porto, Portugal; 2Associate Laboratory for Energy, Transports and Aerospace (LAETA-INEGI), Rua Dr. Roberto Frias 400, 4200-465 Porto, Portugal; 3Algoritmi Research Centre/LASI, School of Engineering, University of Minho, 4800-058 Guimarães, Portugal; 4Department of Textile Engineering, University of Minho, 4800-058 Guimarães, Portugal

**Keywords:** nanomaterials, occupational health and safety, risk management, control banding, standardization, textiles

## Abstract

Nanomaterials offer new technical and commercial opportunities. However, they may also pose risks to consumers and the environment and raise concerns about occupational health and safety. An overview of the standardization in the area of nanomaterials is presented. Focus is given to the standard ISO/TS 12901-2:2014, which describes the use of a control banding approach for controlling the risks associated with occupational exposures to nano-objects and their aggregates and agglomerates greater than 100 nm. The article also presents a case study on a textile finishing company that implements two chemical finishes containing nanomaterials. A risk analysis was conducted to assess the hazards associated with workers handling nanomaterials. Control banding was applied, and measures such as appropriate ventilation and use of protective equipment are proposed to mitigate risks. In some cases, additional measures, such as a closed booth and smoke extractor, are required. The safety data sheets are a primary source of information on how to handle and care for products containing nanomaterials, but the information provided is still limited in terms of the specific hazards and risks posed by nanomaterials.

## 1. Introduction

### 1.1. Nanomaterials

Nanomaterials are incredibly small particles, as tiny as 10,000 times smaller than a human hair. To be considered as “nano”, they must have at least one dimension that is less than 100 nanometers. They can be found in various everyday products, such as food, cosmetics, textiles, electronics, etc. [1,2].

Nanomaterials have unique characteristics that make them very valuable. This is due not only to their miniature size but also factors such as shape and surface area. In reality, the properties of nanomaterials may differ significantly from those of the same materials at larger scales [1,2].

Due to these distinctions in properties, nanomaterials bring new and exciting possibilities to different industries and areas, such as engineering, information technology, medicine, and pharmaceuticals. However, the same characteristics that give nanomaterials their special properties can also lead to potential impacts on human health and the environment [1,2,3,4].

Nanomaterials can be found naturally, such as in volcano emissions, or as a result of human activities, such as diesel exhaust fumes or tobacco smoke. Of particular interest are manufactured nanomaterials designed specifically for a certain use, which are already being incorporated into a vast array of products and applications.

Some nanomaterials have been used for many years, such as synthetic amorphous silica in concrete and food products, while others are more recent discoveries, such as nano-titanium dioxide, used as a UV blocking agent in paints and sunscreens; nano-silver, used as an anti-microbial in textiles and medical applications; or carbon nanotubes, which are used for their mechanical strength, light weight, heat dissipation properties, and electrical conductivity in fields such as electronics, energy storage, spacecraft and vehicle structures, and sports equipment. The market for nanomaterials is rapidly growing and new generations of nanomaterials are being developed at a fast pace. Note that despite the numerous advantages of nanomaterials, there is still a significant amount of knowledge missing regarding their potential health risks [1,4].

In fact, there are significant concerns regarding the health effects of nanomaterials [4]. The Scientific Committee on Emerging and Newly Identified Health Risks (SCENIHR) found that there are proven health hazards associated with a number of manufactured nanomaterials. Not all nanomaterials necessarily have a toxic effect, however, and a case-by-case approach is necessary while ongoing research continues [2].

The most significant effects of nanomaterials have been observed in the lungs, including inflammation, tissue damage, fibrosis, and the generation of tumors. The cardiovascular system may also be affected. Some types of carbon nanotubes have been shown to have effects similar to asbestos. Nanomaterials have been found to reach other organs and tissues, such as the liver, kidneys, heart, brain, skeleton, and soft tissues. The small size and large surface area of particulate nanomaterials in powder form can pose a risk of explosion, while their coarser materials may not [2,5].

Workers can be exposed to nanomaterials in various work environments where nanomaterials are used, handled, or processed, which can cause them to become airborne and potentially inhaled or come into contact with the skin. This type of exposure is most common during the production stage, but workers throughout the supply chain may also come into contact with nanomaterials without realizing it [4]. This raises concerns about the lack of measures in place to prevent exposure, making it crucial to educate workers about the potential risks. The European Agency for Safety and Health at Work (EU-OSHA) has published several resources on risk awareness and communication related to nanomaterials in the workplace [1].

### 1.2. Legislation and Standardization

It is important to emphasize that, in 2014, the European Commission published a recommendation regarding the definition of nanomaterials, which has recently been revised (Commission Recommendation of 10 June 2022):

“Nanomaterial means a natural, incidental or manufactured material consisting of solid particles that are present, either on their own or as identifiable constituent particles in aggregates or agglomerates, and where 50 % or more of these particles in the number-based size distribution fulfil at least one of the following conditions: (a) one or more external dimensions of the particle are in the size range 1 nm to 100 nm; (b) the particle has an elongated shape, such as a rod, fiber or tube, where two external dimensions are smaller than 1 nm and the other dimension is larger than 100 nm; and (c) the particle has a plate-like shape, where one external dimension is smaller than 1 nm and the other dimensions are larger than 100 nm” [2].

Within the European Union, there is legislation in place to protect workers from the potential risks associated with nanomaterials, even though the legislation does not explicitly mention these materials. The Framework Directive 89/391/EEC, the Chemical Agent Directive 98/24/EC, and the Carcinogen and Mutagen Directive 2004/37/EC, as well as the regulations on chemicals such as Registration, Evaluation, Authorization and Restriction of Chemicals (REACH) and Classification, Labelling and Packaging (CLP), are particularly relevant in this context. This means that employers must assess and manage the risks of nanomaterials in the workplace. 

If the use and generation of nanomaterials cannot be eliminated or substituted by less hazardous materials and processes, the exposure of workers must be minimized through preventive measures following a hierarchy of control that prioritizes elimination, substitution, engineering controls, administrative controls, and personal protective equipment in that order. Although many uncertainties still exist, there is a high level of concern about the safety and health hazards of nanomaterials. Therefore, employers and workers must take a precautionary approach to risk management and the selection of prevention measures [1,4].

Standard documents are essential to support the effective implementation of legislation. Standards cover terminology, test methods, material specifications, management systems, and other relevant areas. The authors of this text recently published a review of standards related to the occupational risk and safety of nanotechnologies [6]. A summary and update of this review is presented below.

At the international level, ISO has been developing a large set of standards related to nanotechnology, especially within ISO Technical Committee 229, created in 2005. Working group three deals specifically with health, safety, and environmental aspects of nanotechnologies.

At the European level, the Technical Committee CEN TC 352, created in 2006, is engaged in standardization in the field of nanotechnologies. The standards that have already been published or are under preparation include those that deal with science-based health, safety, and environmental practices.

The European Commission has recognized the importance of standards in supporting legislation and has issued a mandate to the European standardization bodies to develop testing methods and tools for characterizing, understanding the behavior of, and assessing the exposure to nanomaterials. This exposure assessment takes into account the health and safety of workers, as well as the protection of the consumers and of the environment [5].

The coordination of this mandate falls under CEN/TC 352, but multiple CEN and ISO technical committees are involved in its execution. Several European standards have already been published under this mandate, and others are currently being developed. Many of these standards are related to occupational health and safety.

Up to now (April 2023), 32 European standards have been published, including 26 EN/ISO documents developed in conjunction with ISO/TC229. Nine new standards are being prepared. The updated list can be consulted at https://standards.cen.eu/, accessed on 15 April 2023.

Related to ISO/TC229, up to now (February 2023), 102 standard documents have already been published and 32 are under development covering health and safety aspects. The following documents are especially relevant:ISO/TR 12885:2018—Nanotechnologies—Health and safety practices in occupational settings;ISO/TS 12901-1:2012—Nanotechnologies—Occupational risk management applied to engineered nanomaterials—Part 1: Principles and approaches;ISO/TS 12901-2:2014—Nanotechnologies—Occupational risk management applied to engineered nanomaterials—Part 2: Use of the control banding approach;ISO/TR 13121:2011—Nanotechnologies—Nanomaterial risk evaluation;ISO/TR 13329:2012—Nanomaterials—Preparation of material safety data sheet (MSDS).

In 2006, the Organization for Economic Co-operation and Development (OECD) established the Working Party on Manufactured Nanomaterials (WPMN) as a subsidiary body of the OECD Chemicals Committee. This program is focused on examining the implications of manufactured nanomaterials for human health and the environment. Since its inception, the WPMN has published more than 100 documents under the “Safety of Manufactured Nanomaterials” series, some of which are related to standards. A complete list of all freely downloadable documents can be found at: http://www.oecd.org/env/ehs/nanosafety/publications-series-safety-manufactured-nanomaterials.htm, accessed on 15 April 2023 [6].

At the European level, it is also important to mention the Malta Initiative, which was launched during the Maltese EU Council Presidency in 2017. Although this is a self-organized group without any legally binding status, it involves 18 European countries, several Directorates-General of the European Commission, the European Chemicals Agency (ECHA), authorities, research institutions, NGOs, universities, and industry representatives. The goal of this initiative is to make legislation enforceable, particularly in the chemicals sector. To achieve this, it is necessary to ensure that essential test, measurement, and verification procedures are available. The work of the Malta Initiative is focused on amending the OECD Test Guidelines in the area of nanomaterials to ensure that the REACH Regulation is duly adapted to fully cover materials at the nanoscale [6].

### 1.3. Control Banding Approach Applied to Engineered Nanomaterials

The control banding approach has been widely recommended for the selection of exposure controls for engineered nanomaterials. This approach is particularly useful for controlling workplace exposure to potentially hazardous agents with unknown or uncertain toxicological properties and where quantitative exposure estimations are lacking [6,7,8,9].

Already in 2009, the National Institute for Occupational Safety and Health NIOSH) published an extensive review about control banding [4]. There are several control banding tools available, such as NanoSafer, Stoffenmanager-Nano, NanoTool, Precautionary Matrix, ANSES, etc., which can lead to different results [8,9]. In the present study, the ISO Technical Specification ISO/TS 12901-2:2014 was chosen as it is an internationally recognized method.

The control banding process, as defined in ISO/TS 12901-2:2014, involves several elements, including information gathering, assignment of nano-objects to a hazard band based on a comprehensive evaluation of all available data, description of potential exposure characteristics based on workplace scenarios, definition of recommended work environments and handling practices by applying control banding methods, and evaluation of the control strategy or risk banding. Factors such as toxicity, in vivo biopersistence, the ability of particles to reach and be deposited in various regions of the respiratory tract, and their potential to elicit biological responses are considered when assigning a hazard band to each material. Actual exposure measurement data and the potential for dust generation during processes are also taken into account when defining exposure scenarios at the workplace.

A new registration system for nanomaterials has been introduced under REACH Regulation, which took effect in January 2020 (Commission Regulation (EU) 2018/1881 of 3 December 2018). It is recommended that chemical suppliers incorporating nanomaterials provide more information on hazards and risk mitigation measures in safety data sheets based on the guidelines provided in ISO/TR 13329:2012. This information is necessary for the implementation of the control banding approach [10].

More recently, the new Commission Regulation (EU) 2020/878 was introduced on 18 June 2020, which amends Annex II to REACH, providing requirements for compiling safety data sheets used to provide safety information on hazardous chemical substances and mixtures in the EU. This regulation, which came into full force in January 2023, provides more detailed requirements to be included in the safety data sheets of chemicals that include nanoforms.

### 1.4. Use of Nanotechnology in Textiles

Nanotechnology is frequently used in textile products to provide specific functionalities, such as incorporating nanoparticles into textile materials. The common effects include antibacterial effects (using, for instance, nanosilver), ultraviolet protection (using nano-titanium oxide), and self-cleaning through the nanostructuring of the surface. The durability of nanoparticles in textiles depends not only on their attachment to the fabric but also on the impact of the fabric’s lifecycle, which can cause damage to the textile material or the bond between the nanoparticle and fibers due to abrasion, mechanical stress, UV radiation, body fluids, water, detergents, and temperature changes [11,12,13].

Nanoparticles can interact with the human body through inhalation, ingestion, and skin contact, with skin contact being the most relevant pathway for textiles. As part of the European Commission’s mandate, CEN/TC248 (Textile and Textile Products) has developed a specific test method for skin exposure to nanomaterials. The technical report CEN/TR 17222:2019 titled “Textile products and nanotechnologies—Guidance on tests to simulate nanoparticle release—Skin exposure” has been published [14]. This method is based on existing textile standard test methods, which include the extraction of nanomaterials using artificial perspiration solutions under physical stress (a method adapted by Goetz et al. [15]) and measuring the release to the air from the textile due to mechanical action [11,12].

## 2. Materials and Methods

### 2.1. Control Banding: ISO/TS 12901-2:2014

The control banding approach was originally developed by the pharmaceutical industry to safely handle new chemicals with limited or no toxicity data. This practical method can also be utilized to manage the risk of exposure to possibly hazardous agents in the workplace that have unknown or uncertain toxicological properties, such as nanomaterials, where quantitative exposure estimations are not available.

Developing a control banding approach for nanomaterials presents a significant challenge, as it is necessary to determine which parameters and criteria are relevant to assign a nano-object to a control band and which operational strategies to implement. Producers or importers are responsible for identifying whether their product contains nanomaterials and must provide relevant information in safety data sheets, labels, etc., in accordance with existing regulations. By utilizing this information, companies and employees can recognize potential hazards and implement appropriate controls [4,6,11,12,16].

ISO/TS 12901-2:2014 mentioned above is a Technical Specification developed by the ISO that presents a guide to the use of control banding in managing occupational risks associated with engineered nanomaterials. Given the level of uncertainty in assessing potential work-related health risks from nano-objects, including aggregates and agglomerates larger than 100 nm, control banding is a valuable tool for risk assessment and management of nanomaterials. The control banding process outlined in this standard includes several elements, which are well-summarized in the infographic presented in the standard and briefly described below.

First, information must be gathered, and if there are limited or no data available, “reasonable worst-case assumptions” should be used along with management practices appropriate for those options. Then, hazard banding is used to assign a hazard band to nano-objects based on a comprehensive evaluation of available data, taking into account parameters such as toxicity, in vivo biopersistence, and respiratory tract deposition. Exposure banding is used to assign an exposure scenario to an exposure band, taking into account the physical form and amount of the nano-object, dust generation potential, and actual exposure measurement data. Next, control banding is implemented proactively or retroactively to define recommended work environments and handling practices based on hazard banding and fundamental factors mitigating anticipated exposure potential. Finally, periodic and as-needed reviews are conducted to ensure that the information, evaluations, decisions, and actions from the previous steps are kept up to date [4,7,11,12].

ISO/TS 12901-2:2014 describes five hazard bands, as summarized in Table 1 [7].

Details of the allocation of the different hazard bands can be found in Table 1 in the standard.

In terms of exposure banding, the standard proposes four levels from EB 1 to EB 4 corresponding to increased levels of exposure to nanomaterials. In the case of production processes where nanomaterials are handled in liquid form, exposure banding using the levels EB1 or EB2 is normally suggested. When nano-objects are in suspension in a liquid, the choice depends on the amount of liquid and number of nano-objects involved, as well as the potential for aerosol generation [12].

Table 2 presents the control measures proposed in the standard.

Control banding can be determined by combining the hazard bands and the potential exposure band. The corresponding matrix included in ISO/TS 12901-2:2014 is presented in Table 3 [7].

### 2.2. Case Study

In a case study conducted by Ramos et al. [11], the methodology of control banding was applied in a Portuguese textile finishing company specialized in producing knitted fabrics. The study focused on the use of nanomaterials in two chemical finishes—namely, mosquito repellent and antibacterial finish—which were applied to specific customers’ textile products. The risk analysis was mainly focused on four workers who were involved in preparing the finishing baths and operating the stenter frame for knitted fabric finishing [4,7,11,12].

The safety data sheets for the two chemicals, in accordance with the CLP regulation (Regulation EC No 1272/2008), present the hazard statements (H) and precautionary statements (P) shown in Table 4 and Table 5.

The two chemicals, delivered in cans in an aqueous suspension, are applied to cotton knitted fabrics through padding and heat setting in a stenter frame. It should be noted that neither the product information nor the safety data sheets explicitly mention that these chemicals contain nanomaterials, although this information is indirectly provided by the suppliers. In one case, the information just states that the remaining composition of the product is kept secret by the company.

The risk analysis primarily focused on the four workers involved in the preparation of the finishing baths, starting in the chemical warehouse, and in the operation of the stenter frame. The workers’ tasks included opening the cans, weighing the required amount for each batch, transporting the chemicals to the production process, transferring the chemicals to the stenter frame (via automatic dispenser), mixing and preparing the chemicals (with the addition of water), and, finally, developing the finishing process in the stenter.

## 3. Results and Discussion

The hazard band HB B was selected for product A (mosquito repellent) due to the potential for serious eye irritation. Exposure band EB 2 was chosen because it is in a suspension form and used in quantities greater than 1 L, with low potential for aerosol formation. During production, the exposure of workers to nanomaterials is low, resulting in the selection of EB 1. The control band CB 1, which includes natural or mechanical general ventilation, was chosen in accordance with Table 2.

The company has already implemented all of the measures recommended in this study, ISO/TS 12901-2: 2014, and the product safety data sheet for product A. Workers wear protective glasses, 0.7 mm thick butyl rubber gloves, protective clothing, and a respiratory mask when vapors are released. The results of the control banding process for the different tasks performed by workers handling product A are shown in Table 6.

Product A has potential to cause serious eye irritation, so hazard band HB B was chosen for all tasks performed by the workers. Exposure Band EB2 was selected for all tasks except the production phase, as the chemical is in a suspension form with low potential for aerosol formation but is used in quantities greater than 1 L. For the production phase, EB 1 was selected due to the low exposure of workers to the chemicals. In accordance with Table 3, the corresponding control band CB 1 was chosen, needing only natural or mechanical general ventilation [12].

The company has already implemented all suggested measures for product A, including personal protective equipment (PPE), such as protective glasses, gloves, clothing, and respiratory masks.

Table 7 presents the results of the control banding process applied to the different tasks performed by the workers involved in handling product B.

Product B has potential to cause serious eye damage and other health issues. Therefore, the hazard band HB C was selected for all tasks performed by the workers. Exposure band EB 2 was selected since this chemical is in a suspension form with low potential for aerosol formation but is used in quantities greater than 1 L. However, in the case of the production phase, where worker exposure to nanomaterials is low, EB 1 was chosen. In accordance with Table 3, the corresponding control band for product B is CB 3 for all tasks except for the production process, where CB 2 is suggested. CB3 involves the need for enclosed ventilation (a ventilated booth, fume hood, closed reactor with regular opening) whereas, for the production process, local ventilation is sufficient, such as an extractor hood, slot hood, arm hood, table hood, etc.

The company has implemented measures recommended by the product safety data sheet and ISO/TS 12901-2:2014. Workers wear protective glasses, nitrile rubber gloves, protective clothing, and respiratory protection when vapors are released. Further measures requiring more investment will be implemented at a later stage [12].

## 4. Conclusions

The control banding approach was used to suggest risk mitigation measures based on work station analysis and supplier information concerning the chemicals used.

Although there are, in most cases, still no legal occupational exposure limits, it is recommended that the company conduct exposure measurements for chemicals with nanoscale materials at the workplace as these limits can come up in the future.

The lack of information on specific nanomaterials and corresponding risks in safety data sheets is a major issue. To address this, it is suggested that suppliers include more detailed hazard and risk information in safety data sheets based on ISO/TR 13329:2012 recommendations [10]. 

Note that, since January 2023, within the European Union, it is required to include information concerning the nanoforms present in chemicals.

It is also recommended to measure occupational exposure to engineered nanomaterials; for instance, using the method used by Iavicoli et al. [17].

The use of other control banding tools is also suggested for future work, allowing a comparison of the results obtained.

It is worth noting that ISO/TS 12901-2 is currently under revision, so updating the present study in the future is also recommended.

## Figures and Tables

**Table 1 ijerph-20-06011-t001:** Hazard bands.

Category	Hazard
HB A	No significant risk to health
HB B	Slight hazard—slightly toxic
HB C	Moderate hazard
HB D	Serious hazard
HB E	Severe hazard

**Table 2 ijerph-20-06011-t002:** Specific control measures for risk mitigation bands.

Level of Risk	Control Measure
CB 1	Natural or mechanical general ventilation
CB 2	Local ventilation: extractor hood, slot hood, arm hood, table hood, etc.
CB 3	Enclosed ventilation: ventilated booth, fume hood, closed reactor with regular opening
CB 4	Full containment: glove box/bags, continuously closed systems
CB 5	Full containment and review by a specialist: seek expert advice

**Table 3 ijerph-20-06011-t003:** Control band matrix as a result of hazard and exposure potential bands.

Hazard Band	Exposure Band			
	EB 1	EB 2	EB 3	EB 4
HB A	CB 1	CB 1	CB 1	CB 2
HB B	CB 1	CB 1	CB 2	CB 3
HB C	CB 2	CB 3	CB 3	CB 4
HB D	CB 3	CB 4	CB 4	CB 5
HB E	CB 4	CB 5	CB 5	CB 5

**Table 4 ijerph-20-06011-t004:** Product A (mosquito repellent).

Product A (Mosquito Repellent)	
Hazard Statements (H)	Precautionary Statements (P)
H319—causes serious eye irritation	P233—keep container tightly closed P261—avoid breathing dust/fume/gas/mist /vapors/spray
	P305 + P351 + P338—if in eyes, rinse cautiously with water for several minutes; remove contact lenses, if present and easy to do; continue rinsing P305 + P351 + P338—if in eyes, rinse cautiously with water for several minutes; remove contact lenses, if present and easy to do; continue rinsing
	P403 + P233—store in a well-ventilated place, keep container tightly closed
	P501—dispose of contents/container according to local/regional/national/international legislation

**Table 5 ijerph-20-06011-t005:** Product B (antibacterial finish).

Product B (Antibacterial Finish)	
Hazard Statements (H)	Precautionary Statements (P)
H302—harmful if swallowed H318—causes serious eye damageH332—harmful if inhaled	P233—keep container tightly closed P261—avoid breathing dust/fume/gas/mist /vapors/spray P280—wear protective gloves/protective clothing /eye protection/face protection P273—avoid release to the environment
H410—very toxic to aquatic life with long-lasting effects	P301 + P312—if swallowed, call a poison center or doctor/physician if you feel unwell P305 + P351 + P338—if in eyes, rinse cautiously with water for several minutes, remove contact lenses, if present and easy to do, continue rinsing P501—dispose of contents/container to approved incineration unit

**Table 6 ijerph-20-06011-t006:** Selection of control bands for product A (mosquito repellent).

Task	Hazard Band	Exposure Band	Control Band
Opening of the packaging and weighing of the chemicals	HB B	EB 2	CB 1
Transportation and trans-shipment	HB B	EB 2	CB 1
Preparation and start	HB B	EB 2	CB 1
Production	HB B	EB 1	CB 1

**Table 7 ijerph-20-06011-t007:** Selection of control bands for product B (antibacterial finish).

Task	Hazard Band	Exposure Band	Control Band
Opening of the packaging and weighing of the chemicals	HB C	EB 2	CB 3
Transportation and trans-shipment	HB C	EB 2	CB 3
Preparation and start	HB C	EB 2	CB 3
Production (stenter)	HB C	EB 1	CB 2

## Data Availability

Not applicable.

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
