# Peer review of "Managing Nanomaterials in the Workplace by Using the Control Banding Approach†"

_ijerph, 2023, doi:10.3390/ijerph20116011_

Round 1

Reviewer 1 Report

The manuscript described the control banding approach  (ISO/TS 12901-2::2014) to manage the risk of nanomaterials in the workplace. As mentioned in the manuscript, this is an overview of this standard. The authors then described an example of this standard applied in one textile company as a case study and described the preventive measures with comments. Overall, this short review is meaning for researchers interested in establishing safety measures for nanomaterials in the workplace. However, the organization and descriptions of the sections of the materials need much improvement. Here below are detailed comment.

1. The title of the manuscript is too broad for the described materials. Since the “control banding approach” was the focus subject in the manuscript, this term should be used in the title. One title example could be “Managing nanomaterials in the workplace by using the control banding approach”. Another example could be “Control banding approach for managing nanomaterials in the workplace”.

2. The manuscript described only information from the European Commission and at the European level. The authors should add comments on the approaches in other parts of the world in the discussion.

3. The section titles for section 2, “Materials and Method” is not approach. Similar, the “result” section is not appropriate either. The authors are urged to find better titles.

4. Section 2.1 discussed the control banding approach. The authors can improve the description by including a more informative infographics.

5. There are many English grammar errors and mistakes. It is also unusually to have paragraphs with only 2-3 sentences long. The authors are urged to re-organize parts of the manuscript.

6. Out of 14 references, the authors cited three references from their work. The percentage of self-citation is high. The authors should re-work their reference list.

Author Response

Dear Reviewer

Thank you for your comments and suggestions of improvement.

Here are the answers.

  1. The title of the manuscript is too broad for the described materials. Since the “control banding approach” was the focus subject in the manuscript, this term should be used in the title. One title example could be “Managing nanomaterials in the workplace by using the control banding approach”. Another example could be “Control banding approach for managing nanomaterials in the workplace”.

Answer: The title has been revised as suggested.

  1. The manuscript described only information from the European Commission and at the European level. The authors should add comments on the approaches in other parts of the world in the discussion.

Answer: The manuscript has been prepared based on European legislation. But the standards presented are all ISO (International Standards) and reference is also given in terms of work with OECD (International Organization). The reference to EU has been removed in the title of section 1.2.

  1. The section titles for section 2, “Materials and Method” is not approach. Similar, the “result” section is not appropriate either. The authors are urged to find better titles.

Answer: Section titles are basic titles in scientific journals such as Int. J. Environ. Res. Public Health. Note that the control banding has been the method used and the two nanomaterial finishes have been the materials concerned by the study. Section Results contain the results of the control banding evaluation.

  1. Section 2.1 discussed the control banding approach. The authors can improve the description by including a more informative infographics.

Answer: the standard presents in Figure 1 a good summary of the control banding approach. A sentence has been added. Copying the Figure would have copyright problems.

  1. There are many English grammar errors and mistakes. It is also unusually to have paragraphs with only 2-3 sentences long. The authors are urged to re-organize parts of the manuscript.

Answer: the manuscript had been revised by a native English speaker but we understand that some mistakes still remained. We have now corrected several mistakes and we have also reduced the number of sentences in each paragraph.

  1. Out of 14 references, the authors cited three references from their work. The percentage of self-citation is high. The authors should re-work their reference list.

Answer: The three quoted references contain important information that has been essential to prepare this manuscript. But we have also added four additional references, as suggested by another reviewer, so in a total there are now 18 references,

Reviewer 2 Report

ijerph-2268864-comments

The manuscript “Managing nanomaterials in the workplace” provides a description of the need for control banding approaches to mitigate risks of nanomaterials. This is an important topic to cover and therefore I believe this is a worthwhile paper for publication. Although the manuscript is well written and informative, it is lacking slightly in the information provided. In the minor comments below I have tried to cover what I feel these are. My only major concern is the lack of acknowledgement of other control banding approaches. There are a few available and I believe the authors could provide a very useful comparison of these to present how different approaches are provide different (or maybe similar) findings.

Minor comments:

·       The introduction gives a brief account of which nanomaterials are found within industry, but it would be useful to also give an idea of which are produced/used most heavily, e.g. what is the market share of different nanomaterials, and what are the production volumes.

·       Line 104: could the authors expand on the sentence “However, identifying nanomaterials, their emission sources, and exposure levels can be challenging” and identify briefly what these challenges are?

·       Line 119-123: again, could the authors expand of the testing methods and tools they mention? Do these include methods to measure particles in the air?

·       Line 182-184: I believe control banding tools allow entry or promote collection of original data, so the statement “….. information on hazards and risk mitigation measures in safety data sheets, based on the guidelines provided in ISO/TR 13329:2012. This information is necessary for the implementation of the control banding approach” could be edited? E.g. the info on SDS isn’t necessary, but useful, as the user could collect their own data?

·       Section 1.4. Use of nanotechnology in textiles should give an account of which are the main nanomaterials used in the textiles industry and what their functions are.

·       There are a number of tools available that include control banding e.g. CB nanotool and Stoffenmanager nano, these don’t appear to be mentioned?

·       There is very information on the case study material, is this because the company wants this to remain anonymous? If not, could more details be provided? The authors mention that the SDS does not contain much, but the supplier provided more. Having this information would in providing a more meaningful description of their control banding results.

Major comments:

·       The authors have used one control banding approach, although this comes from a very acceptable source, it would be interesting to see whether other control banding tools provide a similar decision/result in their case study. And then could discuss the pros and cons of using different control banding methods.

Author Response

Dear Reviewer

Thank you for your comments and suggestions of improvement.

Here are the answers to your questions.

Minor comments:

  • The introduction gives a brief account of which nanomaterials are found within industry, but it would be useful to also give an idea of which are produced/used most heavily, e.g. what is the market share of different nanomaterials, and what are the production volumes.

Answer: this information can be found in many other scientific papers. The authors did not include It in the present paper as it is not fully relevant for this study.

  • Line 104: could the authors expand on the sentence “However, identifying nanomaterials, their emission sources, and exposure levels can be challenging” and identify briefly what these challenges are?

Answer: this sentence has been removed, as it is not relevant for the paper.

  • Line 119-123: again, could the authors expand of the testing methods and tools they mention? Do these include methods to measure particles in the air?

Answer: in this section it is just mentioned that the EU mandate asks for the development of standard test methods and tools which are at present being developed (see work programme of CEN/TC352 and ISO/TC229. Test methods to measure nanoparticles in the air are being developed in collaboration with CEN/TC137 (ASSESSMENT OF WORKPLACE EXPOSURE).

  • Line 182-184: I believe control banding tools allow entry or promote collection of original data, so the statement “….. information on hazards and risk mitigation measures in safety data sheets, based on the guidelines provided in ISO/TR 13329:2012. This information is necessary for the implementation of the control banding approach” could be edited? E.g. the info on SDS isn’t necessary, but useful, as the user could collect their own data?

Answer: The SDS should contain all relevant information to perform control banding. A sentence concerning the new requirements in the European Union to include information concerning the nanoforms present in the chemicals has been added in the conclusion, as suggested by another reviewer.

  • Section 1.4. Use of nanotechnology in textiles should give an account of which are the main nanomaterials used in the textiles industry and what their functions are.

Answer: A short sentence has been added to address this comment (first paragraph of section 1.4). In reference 8 (now renumbered as reference 12) authors have presented more details about the nanomaterials used in textiles and the functionalities obtained.

  • There are a number of tools available that include control banding e.g. CB nanotool and Stoffenmanager nano, these don’t appear to be mentioned?

Answer: we have added a sentence in section 1.3 mentioning other control banding tools. Also in the conclusion we have added a sentence with a suggestion of future work using other CB tools. Three further references have been added which include other CB tools.

  • There is very information on the case study material, is this because the company wants this to remain anonymous? If not, could more details be provided? The authors mention that the SDS does not contain much, but the supplier provided more. Having this information would in providing a more meaningful description of their control banding results.

Answer: the company and the supplier did not disclose detailed information about the two chemicals. Nevertheless, the SDS contained enough information to classify the two chemicals according to the control banding matrix.

Major comments:

  • The authors have used one control banding approach, although this comes from a very acceptable source, it would be interesting to see whether other control banding tools provide a similar decision/result in their case study. And then could discuss the pros and cons of using different control banding methods.

Answer: the study is all based on the method described in ISO standard, an internationally adopted method. A brief reference to other control banding tools has been included in the introduction and also in the conclusion as a suggestion for future work, also taking into account the publication, in a near future, of a new version of the standard ISO/TS 12901-2.

Reviewer 3 Report

This manuscript provides an overview of the available standards relevant to occupational exposure to nanomaterials, with a special focus on ISO/TS 12901-2:2014 on control banding application for managing the occupational risk of engineered nanomaterials. This is a quite interesting topic but not often explored in scientific papers that is of interest for the journal’s audience. A case study is also presented, where the ISO control banding approach is applied to manage the risks in the textile industry associated with handling of two chemical finishes containing nanomaterials.

Some aspects need clarification and the manuscript can be improved as stated below.

Title

The title is rather general and does not reflect the true content of the manuscript.

Abstract

Page 1, line 14: “This standard presents a practical approach that can be used to manage workplace exposure to hazardous agents with unknown or uncertain toxicological properties, where exposure levels cannot be accurately estimated.”

ISO/TS 12901-2:2014 describes the use of a control banding approach for controlling the risks associated with occupational exposures not to hazardous agents in general but more specifically to nano-objects, and their aggregates and agglomerates greater than 100 nm (NOAA). 

Introduction

Page 1, line 34: In reality, the properties of nanomaterials differ from those of the same materials at larger scales [1,2].

Properties of nanomaterials may or may not differ from the same material in bulk.

The authors must review the list of references used and expand it. For instance, some relevant literature is missing, for instance 2017 WHO guidelines on protecting workers from potential risks of manufactured nanomaterials. Some sentences need to be supported by references. Some examples are provided below:

Page 1, line 30: “To be consider as nano they must have at least one dimension that is less 30 than 100 nanometers.”

Page 1, line 45: “Some nanomaterials have been used for many years, such as synthetic amorphous silica in concrete and food products, while others are more recent discoveries, such as nano-titanium dioxide as a UV blocking agent in paints and sunscreens, nano-silver as an anti-microbial in textiles and medical applications, or carbon nanotubes, which are used for their mechanical strength, light weight, heat dissipation properties, and electrical conductivity in fields such as electronics, energy storage, spacecraft and vehicle structures, and sports equipment.”

Page 2, line 61: “The most significant effects of nanomaterials have been observed in the lungs, including inflammation, tissue damage, fibrosis, and the generation of tumors. The cardiovascular system may also be affected. Some types of carbon nanotubes have been shown to have effects similar to asbestos. Nanomaterials have been found to reach other organs and tissues such as the liver, kidneys, heart, brain, skeleton, and soft tissues.”

Page 2, line 68: “Workers can be exposed to nanomaterials in various work environments where nanomaterials are used, handled, or processed, which can cause them to become airborne and potentially inhaled or come into contact with the skin. This type of exposure is most common during the production stage, but workers throughout the supply chain may also come into contact with nanomaterials without realizing it.”

Page 2, line 53: “Note that despite the numerous advantages of nanomaterials, there is still a significant amount of knowledge that is missing regarding their potential health risks [1,4].”

Reference #4 is specific to textile materials, thus not the most adequate to be used to support such a general statement.

Page 3, line 129: “Until now (February 2023), 29 European standards have been published, including 18 EN/ISO documents, developed in conjunction with ISO/TC229. 11 new standards are being prepared. The updated list can be consulted at https://standards.cen.eu/.”

Please update the provided information.

Much of the information provided in the Material and Methods section is rather general, not specific to the selected case study, and can be moved to the Introduction subsection 1.3 on Control banding approach applied to engineered nanomaterials. Since control banding is one of the focus of the manuscript, it would also be nice to include in 1.3 subsection a brief overview of other available control banding tools specific to engineered nanomaterials (e.g. CB NanoTool, Stoffenmanager Nano, NanoSafer), as well as on their similarities and differences in features such as their scope and applicability, parameters used in the hazard/severity, banding and exposure/probability/emission potential banding, and in the classification of risk or control bands.

ISO/TS 12901-2:2014 describes five hazard categories; Category E (severe hazard) is missing in Table 1. Besides, Table 1 is a very simplified version of the original one, which makes reader’s understanding of hazard band selection for Products A and B difficult. I would suggest moving Tables 1 and 3 (Tables 4 and 5, too) to the Supplementary Materials section.

Tables 2 and 3 have the same caption.

Page 6, line 257: “The study focused on the use of nanomaterials in two chemical finishes, namely mosquito repellent and antibacterial finish, which were applied to specific customers' textile products.”

“It should be noted that neither of the product information or safety data sheets explicitly mention that these chemicals contain nanomaterials, although this information is indirectly provided by the suppliers. In one case, the information just states that the remaining composition of the product is kept secret by the company.”

None of the nanomaterials included in the selected finishes products A (a mosquito repellent) and B (antibacterial) is identified. What do authors mean by “It should be noted that neither of the product information or safety data sheets explicitly mention that these chemicals contain nanomaterials, although this information is indirectly provided by the suppliers.

If so, how can one ascertain the presence of nanomaterials (and of what type) in the selected products? What do the authors mean by “indirectly provided by the suppliers”?

Conclusions

Page 8, line 312: “The company is recommended to conduct exposure measurements to chemicals with nano-scale materials at the workplace and compare them to normative and legal limits.”

This statement might lead one to think that occupational exposure limits (OELs) to nanomaterials exist. In fact, it would be important to address this topic in the manuscript, i.e. current lack of OELs specific for nanomaterials. 

Page 8, line 327: “It is also recommended to measure occupational exposure to engineered nanomaterials using methods like the one used by Iavicoli et al. [14].”

Please elaborate on this.

Author Response

Dear Reviewer

Thank you for your comments and suggestions of improvement.

Here are the answers to your questions.

Title

The title is rather general and does not reflect the true content of the manuscript.

Answer: the title has been adapted also according to the suggestion of another reviewer.

Abstract

Page 1, line 14: “This standard presents a practical approach that can be used to manage workplace exposure to hazardous agents with unknown or uncertain toxicological properties, where exposure levels cannot be accurately estimated.”

ISO/TS 12901-2:2014 describes the use of a control banding approach for controlling the risks associated with occupational exposures not to hazardous agents in general but more specifically to nano-objects, and their aggregates and agglomerates greater than 100 nm (NOAA).

Answer: the sentence included in the abstract has been revised accordingly, also taking into account that there are severe restrictions in terms of number of words in the abstract.

Introduction

Page 1, line 34: In reality, the properties of nanomaterials differ from those of the same materials at larger scales [1,2].

Properties of nanomaterials may or may not differ from the same material in bulk.

Answer: The properties of nanomaterials are normally significantly different from non nano equivalent, namely in terms of physical properties. The wording “may differ significantly” has been added.

The authors must review the list of references used and expand it. For instance, some relevant literature is missing, for instance 2017 WHO guidelines on protecting workers from potential risks of manufactured nanomaterials. Some sentences need to be supported by references. Some examples are provided below:

WHO guidelines on protecting workers from potential risks of manufactured nanomaterials

2 February 2017

Answer: Thank you for your suggestion. This important reference has been added as new [4] and quoted in the introduction.

Page 1, line 30: “To be consider as nano they must have at least one dimension that is less 30 than 100 nanometers.”

Answer: References 1 and now also 2 are quoted.

Page 1, line 45: “Some nanomaterials have been used for many years, such as synthetic amorphous silica in concrete and food products, while others are more recent discoveries, such as nano-titanium dioxide as a UV blocking agent in paints and sunscreens, nano-silver as an anti-microbial in textiles and medical applications, or carbon nanotubes, which are used for their mechanical strength, light weight, heat dissipation properties, and electrical conductivity in fields such as electronics, energy storage, spacecraft and vehicle structures, and sports equipment.”

Answer: at the end of this paragraph the references 1 and 4 are quoted.

Page 2, line 61: “The most significant effects of nanomaterials have been observed in the lungs, including inflammation, tissue damage, fibrosis, and the generation of tumors. The cardiovascular system may also be affected. Some types of carbon nanotubes have been shown to have effects similar to asbestos. Nanomaterials have been found to reach other organs and tissues such as the liver, kidneys, heart, brain, skeleton, and soft tissues.”

Answer: at the end of the paragraph the references 2 and 5 are quoted.

Page 2, line 68: “Workers can be exposed to nanomaterials in various work environments where nanomaterials are used, handled, or processed, which can cause them to become airborne and potentially inhaled or come into contact with the skin. This type of exposure is most common during the production stage, but workers throughout the supply chain may also come into contact with nanomaterials without realizing it.”

Answer: References 4 has been added also here. Note that although no specific question is put in this sentence, for instance in the textile and clothing chain workers involved in the clothing manufacturing are not aware of the presence of nanomaterials in the textiles. Same concerning workers involved in the distribution chain.

Page 2, line 53: “Note that despite the numerous advantages of nanomaterials, there is still a significant amount of knowledge that is missing regarding their potential health risks [1,4].”

Reference #4is specific to textile materials, thus not the most adequate to be used to support such a general statement.

Answer: Reference 4, now renumbered 5 (concerning Chapter VII- Governance/Legislation/EU legal framework-REACH; Occupational health aspects (EU) of the book) includes a general survey of legal aspects concerning nanomaterials, not specifically related to textiles.

Page 3, line 129: “Until now (February 2023), 29 European standards have been published, including 18 EN/ISO documents, developed in conjunction with ISO/TC229. 11 new standards are being prepared. The updated list can be consulted at https://standards.cen.eu/.”

Please update the provided information.

Answer: Information has been updated with data in April 2023. 3 more standard documents have been published, also the number of EN/ISO documents was not correct. Data on ISO/TC 229 has also been updated.

Much of the information provided in the Material and Methods section is rather general, not specific to the selected case study, and can be moved to the Introduction subsection 1.3 on Control banding approach applied to engineered nanomaterials. Since control banding is one of the focus of the manuscript, it would also be nice to include in 1.3 subsection a brief overview of other available control banding tools specific to engineered nanomaterials (e.g. CB NanoTool, Stoffenmanager Nano, NanoSafer), as well as on their similarities and differences in features such as their scope and applicability, parameters used in the hazard/severity, banding and exposure/probability/emission potential banding, and in the classification of risk or control bands.

Answer: the study is all based on the method described in ISO standard, an internationally recognized method. A brief reference to other control banding tools has been added in the introduction and also in the conclusion, as also suggested by another reviewer. Three supplementary references have been added in which comparison of different CB tools can be found.

ISO/TS 12901-2:2014 describes five hazard categories; Category E (severe hazard) is missing in Table 1. Besides, Table 1 is a very simplified version of the original one, which makes reader’s understanding of hazard band selection for Products A and B difficult. I would suggest moving Tables 1 and 3 (Tables 4 and 5, too) to the Supplementary Materials section.

Answer: in fact, the fifth hazard band (severe hazard) is missing in table 1. We have also added the following sentence: Details of the allocation of the different hazard bands can be found on Table 1 of the standard.

Tables 2 and 3 have the same caption.

Answer: thank you for your remark. Caption of Table 3 has been corrected.

Page 6, line 257: “The study focused on the use of nanomaterials in two chemical finishes, namely mosquito repellent and antibacterial finish, which were applied to specific customers' textile products.”

“It should be noted that neither of the product information or safety data sheets explicitly mention that these chemicals contain nanomaterials, although this information is indirectly provided by the suppliers. In one case, the information just states that the remaining composition of the product is kept secret by the company.”

None of the nanomaterials included in the selected finishes products A (a mosquito repellent) and B (antibacterial) is identified. What do authors mean by “It should be noted that neither of the product information or safety data sheets explicitly mention that these chemicals contain nanomaterials, although this information is indirectly provided by the suppliers.

If so, how can one ascertain the presence of nanomaterials (and of what type) in the selected products? What do the authors mean by “indirectly provided by the suppliers”?

Answer: although the safety data sheets do not directly refer to the content of nanomaterials in the chemical finishes, the suppliers in commercial contacts claim that the two finishes are based on nanotechnology. Literature also supports that these effects are now obtained by the use of nanomaterials. The fact that suppliers did not want to specifically mention this fact in the safety data sheets is probably due to avoid eventual legal problems (teher are several restrictions on the use of nanomaterials in textiles, for instance in GOTS (Global Organic Textile Standard). Note that, as mentioned in the paper, since January 2023 within the European Union it is required that this information in included in the safety data sheets. A sentence has been added in the conclusions to highlight this fact.

Conclusions

Page 8, line 312: “The company is recommended to conduct exposure measurements to chemicals with nano-scale materials at the workplace and compare them to normative and legal limits.”

This statement might lead one to think that occupational exposure limits (OELs) to nanomaterials exist. In fact, it would be important to address this topic in the manuscript, i.e. current lack of OELs specific for nanomaterials.

Answer: this sentence in the conclusion has been rewritten according to the suggestion.

Page 8, line 327: “It is also recommended to measure occupational exposure to engineered nanomaterials using methods like the one used by Iavicoli et al. [14].”

Please elaborate on this.

Answer: the phrase has been rewritten

Round 2

Reviewer 1 Report

The authors answer some of the questions from the two reviews. There are still lots of grammatical errors and they are not corrected. The authors even mention "Figure 1" which does not exist in the manuscript. The authors cited an example which was already published. So they should say that this is a review of a past exam. Overall, the authors should spend some time to examine the writing carefully.

Author Response

Dear Reviewer

Thank you for your comments and suggestions of improvement.

Here are the answers to your questions.

As stated in the previous answer, we have had the text revised by a native English speaker. We asked him to check again all the text and we have made some improvements, duly marked in the text. Note that there were not ”a lot” of grammatical errors, but we tried to make some sentences more clear.

Concerning Figure 1, we have mentioned that this Figure presents a good infographic about the control banding method. This text has been added according to your suggestion. We have tried now to better clarify this sentence. As we have told, we have avoided to copy this Figure in our paper due to copyright restrictions.

The previously published work is duly quoted.

Reviewer 2 Report

·         The revision appears to have been minimal, and not enough to justify publication. The main emission here still remains the lack of comparison to other approaches of control banding; to give this publication some critical views and novel conclusions it would have been necessary to rate the ISO standard method against others available, to give an opinion of whether the ISO standard adopted method is satisfactory or whether it has gaps/limitations that could be supported by other methods.

Author Response

Dear Reviewer

Thank you for your comment and suggestion of improvement.

Here is the answer to your comment.

As we have stated and better clarified in the revised manuscript, we have used the control banding method in an ISO standard, an internationally adopted method. Note that this method has been developed taking into account previously existing methods. ISO/TS 12901-2 has been developed at international level on the basis of work done by experts from several countries and, as an ISO standard, it is recognized all over the world (note that ISO has a total of 215 member countries).

According to your suggestion, we have made reference to other control banding methods (we have added 4 references related to this topic – numbers 4, 8, 9 and 10).

We have now added in the conclusion two sentences with a suggestion of future work using other CB tools, also taking into account the publication, in a near future, of a new version of the standard ISO/TS 12901-2 (although mentioned in the answer we had sent you before, these sentences were missing in the previous revision of the manuscript).